# Lightweight Detection Network Based on Sub-Pixel Convolution and Objectness-Aware Structure for UAV Images

**DOI:** 10.3390/s21165656

**Published:** 2021-08-22

**Authors:** Xuanye Li, Hongguang Li, Yalong Jiang, Meng Wang

**Affiliations:** 1School of Electrical and Information Engineering, Beihang University, Beijing 100191, China; lixuanye1902@buaa.edu.cn (X.L.); yukata@buaa.edu.cn (M.W.); 2Unmanned System Research Institute, Beihang University, Beijing 100191, China; allenyljiang@buaa.edu.cn

**Keywords:** lightweight convolutional neural network, object detection, UAV images

## Abstract

Unmanned Aerial Vehicles (UAVs) can serve as an ideal mobile platform in various situations. Real-time object detection with on-board apparatus provides drones with increased flexibility as well as a higher intelligence level. In order to achieve good detection results in UAV images with complex ground scenes, small object size and high object density, most of the previous work introduced models with higher computational burdens, making deployment on mobile platforms more difficult.This paper puts forward a lightweight object detection framework. Besides being anchor-free, the framework is based on a lightweight backbone and a simultaneous up-sampling and detection module to form a more efficient detection architecture. Meanwhile, we add an objectness branch to assist the multi-class center point prediction, which notably improves the detection accuracy and only takes up very little computing resources. The results of the experiment indicate that the computational cost of this paper is 92.78% lower than the CenterNet with ResNet18 backbone, and the mAP is 2.8 points higher on the Visdrone-2018-VID dataset. A frame rate of about 220 FPS is achieved. Additionally, we perform ablation experiments to check on the validity of each part, and the method we propose is compared with other representative lightweight object detection methods on UAV image datasets.

## 1. Introduction

With the advance of UAV technology and the growth of UAV suppliers, UAVs are becoming more cost-efficient. Meanwhile, due to their mobility, being autonomous, and their processing capabilities, UAVs are considered in many intelligent transportation system (ITS) application domains [1], such as traffic state estimation, traffic control, incidence emergency response and so on. Compared to fixed road monitoring devices, using UAV cameras for traffic monitoring has the following advantages [2,3]: (1) UAVs have wider spatial coverage; (2) UAVs are easier to maintain; and (3) UAVs are more cost-efficient. Modeling traffic flow to evaluate traffic conditions is a significant part of ITS [4]. The detection of objects of interest from UAV images/videos is the initialization process of traffic state estimation [5], which provides fast and accurate traffic data collection.

Most UAV visible image object detection algorithms are based on widely used and universally structured methods, such as Faster RCNN or SSD, which target the small scale and dense distribution of UAV image objects, either by complicating the network structure or the detection process [6,7,8,9], or by introducing novel ways of data augmentation [7,10], ultimately making the algorithms perform well on UAV datasets. Typically, the optimisation goal of these algorithms is to improve accuracy as much as possible, with less consideration given to efficiency, and the few fast algorithms are only somewhat faster relative to their predecessors, falling far short of the standard of real-time. Online object detection based on UAV onboard platforms is of great importance, not only to improve the flexibility of UAV applications and the intelligence of the UAV itself, but also to overcome the harsh communication environment in order to work. However, the storage and computational resources of the UAV onboard platform are limited, requiring algorithms with low computational and parametric quantities.

Targeted at real-time on-board platforms with visible-light cameras, this paper develops an efficient object detection method with less computation and fewer parameters. In our work, a lightweight detection network is proposed to predict object centers. Firstly, a backbone is developed by removing the last convolutional layer of the MobileNetV2 [11] network. Then, we provide an analysis of the computational burdens in different parts of the existing detection models and propose an efficient detection head, which implements upsampling and detection together using one layer. To achieve both functions while reducing computational burdens, we share the convolutional layer of sub-pixel convolution [12] and detection head to form a unified module. Besides, we add a novel objectness branch to the detection head. Whether a point on the characteristic spectrum is the center point of an interested object is determined by the binary classification result provided by the objectness branch. Compared with the CenterNet, which has a ResNet18 backbone, our proposed model reduces the computational cost by 92.78%, reduces the parameter size by 86.73%, and improves the mAP by 2.8 points on the UAV image dataset. The influence of each part on the computational cost, parameter size and detection precision are demonstrated by an extensive ablation study.

This paper has contributed to the existing research in the following three ways:(1)The paper proposes a lightweight as well as anchor-free framework for UAV images, which efficiently reduces the time of computation and the memory consumption. The framework fits better for performing real-time detection in resource-constrained scenarios.(2)We introduce sub-pixel convolution to the small object detection and draw the support of a sub-pixel convolution structure to develop a simultaneous upsampling and detection module. The module implements upsampling and detection together using one convolutional layer that improves the efficiency and without reducing detection accuracy.(3)We add a novel objectness branch to the detection head. The additional supervision in the form of objectness makes the model develop more robust feature representations and perform better in detection.

The structure of this paper is designed as follows. The second section performs a literature review, the third section introduces the methods employed in detail, the fourth section advances the evaluation metric and dataset, and records the experimental setups and experimental results, while the last section presents a conclusion and puts forward a few directions for future research.

## 2. Related Work

In this part, the CNN-based object detection approaches and previous work on object detection in UAV images are introduced.

### 2.1. General Object Detection

Most of the existing CNN-based object detection approaches take advantage of anchor boxes, including single-stage and two-stage approaches. For two-stage approaches, the former stage generates an exiguous set of candidate regions relying on the anchor boxes and the second stage performs classification and regression on the candidate regions. Typical two-stage methods include Faster RCNN [13], RFCN [14], Cascade RCNN [15], FPN [16], and so forth. In single-stage methods, the anchor boxes that are densely sampled from the feature map are directly classified and regressed, and the region proposal network (RPN) is omitted. Typical single-stage methods include SSD [17], YOLOv3 [18], RetinaNet [19], DSOD [20], RefineDet [21], and so forth. Two-stage methods achieved a higher accuracy while suffering from larger computational complexity and higher memory cost due to its inherent structure. Single-stage methods generally achieve a tradeoff between detection speed and accuracy. For instance, Faster RCNN with a VGG16 backbone network runs on the graphics processing unit (GPU) at about seven frames per second (FPS), while the SSD can reach 46 FPS.

Due to the gap between the computational resources in GPUs and those in embedded platforms, achieving real-time performance on mobile platforms is extremely hard using the above-mentioned approaches. Therefore, some lightweight detection networks are proposed, most of which are based on the single-stage methods. Based on SSD, Pelee [22] used a more efficient PeleeNet backbone, which drew on the idea of DenseNet [23], and the number of anchor boxes was reduced at the same time. Finally, it came to a very lightweight model for scenarios with limited hardware resources. Tiny SSD [24] and Tiny YOLOv3 [18] also introduced lightweight backbone networks to reduce computation. The backbones of Tiny YOLOv3 and Tiny SSD are DarkNet19 and SqueezeNet [25], respectively. In [26], Gabor filters with fewer filter parameters to learn are incorporated into the convolution filter to improve the robustness of DCNNs; it may also perform as well as a lightweight backbone for the detection network. YOLO-LITE [27] was especially developed for GPU-free devices. Even more efficient than YOLOv2, the method reaches 20 FPS on GPU-free computers. Tiny DSOD [28] is the lightweight implementation of DSOD, it introduced depth separable convolutions into the DenseNet-like backbone network and the feature pyramid network (FPN). Basically, most of the lightweight object detection networks make their structure more lightweight and effective by reasonably simplifying backbone networks and detection heads.

In recent years, a number of anchor-free object detection approaches have emerged. The newly emerged methods abandoned the anchorage box; as a result, the algorithm changes from box-based classification and regression to corner point based [29], center point based [30], or keypoint based [31] ones. CornerNet [29] forecasts the upper left as well as the bottom right corners of an object’s bounding box, and utilizes associative embedding to group the corners of the same target to finish the detection of an object. CenterNet [30] borrows part of CornerNet’s ideas, and its solution is more intuitive: Detection is done by forecasting the object’s focus and scale as well as by eliminating the process of corner matching. Meanwhile, the detection accuracy is higher. Compared with anchor-based approaches whose detection performance is affected by the settings of anchors, anchor-free methods are more robust. Moreover, anchor-free methods are mostly composed of a single stage, and the detection accuracy is comparable to the anchor-based and single-stage methods. It has a broad development prospect at present. At the same time, some generic modules were introduced to detection networks. For example, to overcome the confusion of background and objects, an Inference-aware Feature Filtering (IFF) approach [32] was proposed, which optimizes feature learning in a theoretical framework by introducing a feedback architecture in either anchor-based or anchor-free detection networks.

### 2.2. Object Detection in UAV Images

Because of complex ground scenes and small object size, common CNN-based object detection methods cannot achieve satisfactory performance on UAV images. To achieve better performance, most existing object detection methods adopt large-scale classification networks (VGG [33], ResNet101 [34]) as a backbone to extract features and add optimized single-stage or two-stage detection heads.

The method in [6] introduced DeForm convolutional layers within the backbone and proposed an interleaved cascade architecture. Meanwhile, multi-model fusion was used to deal with class imbalance problems. On the basis of Faster RCNN, Reference [35] proposed a coupled R-CNN network to detect the vehicle. The task combined an accurate-vehicle-proposal-network (AVPN) with a vehicle property learning network in order to predict the spot and attributes of the vehicle synchronously. Reference [36] proposed a depthwise separable attention-guided network (DAGN), which integrated the feature series with a concentration block to make sure that the model is able to brilliantly differentiate significant and trivial features. Reference [37] integrated the overall and partial fusion strategy with a progressive network with varying scales to fullfill detection in a more accurate manner. In [7], an anchor-free method was introduced. Compared to the typical method based on center point prediction, the scale of the object needs to be regressed twice to obtain a more accurate bounding box. The method in [8] used enhanced SSD to detect vehicles in drone images to assist vehicle counting and to tackle the traffic density estimation tasks. In [9], a mask resampling module (MRM) was constructed to boost the unbalanced datasets. Besides, a coarse anchor-free detector (CPEN) and a fine anchor-free detector (FPEN) were adopted to forecast the focuses of the small object flocks and to locate the locations of small objects in a valid and accurate manner. In [38], a parallel lightweight auxiliary meshwork and an ovonic network were proposed to effectively process the semantic information from low to high levels. This method considered the accuracy and efficiency comprehensively, and finally reached 91 FPS on the GPU. But it is still far from the speed of the lightweight detection methods with a frame rate of over 100 FPS.

In summary, a lot of work has provided solutions for accurate UAV image object detection. Nevertheless, the above approaches are generally grounded on GPU platforms, which makes it difficult to apply them to mobile terminals. Current research on lightweight networks under resource-constrained scenarios still mainly focuses on non-UAV images.

## 3. Materials and Methods

The proposed framework is targeted at deployment on resource-constrained on-board platforms. The overall structure of the lightweight object detection network is proposed with the consideration of both speed and accuracy. The structure is shown in Figure 1 and Table 1. The following subsections will further expand on the details of each part.

### 3.1. Overall Architecture

Our work no longer uses anchor boxes to indicate objects, but instead predicts the center point and the scale of an object of interest.

As is shown in Figure 1, our framework uses a single-scale structure. The input image undergoes a 32× down-sampling after it is processed by the revised MobileNetV2 backbone network. Different from anchor-based methods, which densely generate multiple boxes for each pixel in feature maps, our method conducts sparse sampling on feature maps. If the feature map with a large receptive field is directly fed into the detection head, the sampling will be too sparse, and is not conducive to the detection of objects. Consequently, the feature maps output by the backbone meshwork need to be up-sampled before being fed into the subsequent detection head network. We propose a simultaneous up-sampling and detection module that conducts up-sampling and detection simultaneously. The up-sampling and detection functions are based on a unitary 1 × 1 convolutional layer.

The structure of the detection head involves four branches, which are the multi-classification branch, objectness branch, bounding box scale branch, and center point offset branch. The outputs of the four branches are in same spatial size (4× down-sampling of input), the only distinction is how many output channels each branch has. The multi-classification branch generates a series of heatmaps corresponding to the number of interested categories. The heatmaps indicate the probability of each pixel as the center point of each interested categories.

The output of the bounding box scale branch indicates the width and height of the objects corresponding to center points. The output of the center point offset branch indicates the coordinate compensation of center points to correct the discretization error brought by the down-sampling. The above three branches can generate bounding boxes with confidence scores. In addition, we added an objectness branch, which generates a heatmap indicating whether a certain pixel in space corresponds to the center of an interested object. The gradients introduced by the objectness branch make the model develop more robust feature representations and thus perform better in detection.

The training is conducted in an end-to-end process. During inference, the original picture is scaled before being fed to the detection network, and the heatmaps of the object center points with specific scale and offset values are output. Our framework uses the post-processing method of extracting the peak in the local area of the center point heatmap for deduplication instead of non-maximum suppression (NMS). The value of a pixel is kept unchanged if it is the maximum of eight nearest neighbors, and is set to 0 otherwise. In practice, it can be easily achieved by maxpooling, and the computation complexity is less than NMS.

### 3.2. Lightweight Feature Extractor

CenterNet has achieved impressive accuracy and real-time performance on GPU. Its most efficient version uses ResNet18 as the backbone and the inference speed can reach 140 FPS. However, the computational cost and the number of learnable parameters can still be reduced. As shown in Table 2, the ResNet18 feature extractor and the transpose convolution operation occupy a large proportion of the computational cost and parameter size.

Like most lightweight detection networks, we replaced the backbone taken from the classification network with a revised MobileNetV2. MobileNetV2 adopts the deeply demountable convolution and inverted residual structure, a mobile network structure that has been far and wide accepted in tasks such as detection and division.

Depthwise separable convolution factorizes standard convolution to a deep one as well as a pointwise one. The computational cost of standard convolution is as follows:(1)$$N_{s}=K\times K\times C_{in}\times C_{out}\times F_{out}\times F_{out},$$
where *K* is the measurement of kernel, $C_{in}$ is the amount of input ends, $C_{out}$ is the amount of output ends, $F_{out}$ is the spatial size of the output. The computational costs of deep convolution $N_{dw}$ and pointwise convolution $N_{pw}$ are:(2)$$\begin{array}{cc} N_{dw} & =K\times K\times C_{in}\times F_{out}\times F_{out}, \end{array}$$(3)$$\begin{array}{cc} N_{pw} & =C_{in}\times C_{out}\times F_{out}\times F_{out}. \end{array}$$

Compared with standard convolution, deeply separable convolution cuts the computational cost down to:(4)$$\frac{N_{dw}+N_{pw}}{N_{s}}=\frac{1}{C_{out}}+\frac{1}{k^{2}}.$$

Generally, the structure in front of the last pooling layer of the classification network is chosen to be the backbone to extract features. Through experiments, we find that if we take MobileNetV2 as a part of the backbone, the final 1 × 1 convolutional layer has a negative effect on detection accuracy. The reason lies in the fact that MobileNetV2 is a structure designed for classification tasks and the ultimate goal is to obtain well-discriminated feature vectors which are then put to fully-linkedl ayers to be classified. The non-linear RELU function following the last 1 × 1 convolutional layer may spoil the informative features output by the last linear residual block.

At the same time, the last 1 × 1 convolutional layer outputs quite a large dimension size (1280 dimensions), which also brings a huge computational burden to the sub-sequent up-sampling operation and detection head, so it is extremely beneficial to remove. If the layer is removed, the output dimension of the backbone is cut to 1/4 of the original (320 dimensions). In our proposed network, the computational cost of the sub-sequent structure is also reduced to 1/4.

Considering that the transpose convolution for up-sampling takes up a lot of resources, we replace the transpose convolution with the sub-pixel convolution and try to construct a simultaneous up-sampling and detection module. See the next section for details.

### 3.3. Simultaneous Up-Sampling and Detection Module

As a common method for up-sampling, transpose convolution is different from interpolation or up-pooling. The advantage is that transpose convolution is learnable and can make the results more refined. However, transpose convolution will produce a checkerboard effect [39] at a certain stride and kernel size (for example, when the stride is two and the kernel size is odd), and the up-sampling performance of transpose convolution is closely related to convolutional kernel size. These lead to the necessity of large kernel sizes. As a result, transpose convolution takes up 26% of the computation and parameters of the entire CenterNet.

We use sub-pixel convolution instead of transpose convolution. Sub-pixel convolution is also a learning-based up-sampling method. It can be defined as:(5)$$FM^{HR}=PS\left(W_{L}\times FM^{LR}+b_{L}\right),$$
where PS replumes a low-resolution $H\;\times \;W\;\times \;C\cdot r^{2}$ feature map to a high-resolution feature map $FM^{HR}$ with a shape of rH×rW×C as a periodic shuffling operator. The $W_{L}$ and $b_{L}$ are convolution operators that are used to raise the dimension of the original low-resolution map $FM^{LR}$ to $r^{2}$ times as large. In brief, the convolutional layer is first used to raise the dimension of input, and then the convolutional layer output is rearranged by the periodic shuffling to obtain the result of up-sampling. Since sub-pixel convolution and transpose convolution have different principles, the sub-pixel convolution will not be affected by the checkerboard effect. At the same time, sub-pixel convolution and the detection head have convolution structures; we try to share the structure of these two parts. Our simultaneous up-sampling and detection module and its counterparts in CenterNet are shown in Figure 2.

In practice, the upsampled feature map will serve as the input to each branch in the detection head. Each branch contains a 1 × 1 convolutional layer, the output of which is $8^{2}$ times the final output dimensions. Finally, the prediction results of the corresponding branch is obtained after periodic shuffling. As shown in Figure 1, simultaneous upsampling and detection module takes low-resolution 16 × 16 feature maps as the input and outputs 128 × 128 prediction heatmaps. This structure takes advantage of the characteristics of sub-pixel convolution; it not only reduces computation in comparison to the transpose convolution operation, but also simplifies the detection head by integrating the up-sampling operation into the detection head to further reduce the computational burden. Our experiments demonstrate that this structure only reduces the amount of computation and parameters without influencing accuracy.

### 3.4. Objectness Branch

The primary motivation of this work is to obtain an online real-time object detection network with no resource constraints. Therefore, we have introduced a lightweight backbone network as well as a detection device. To avoid the drop in detection accuracy brought by applying a lightweight structure and the characteristic of UAV images, an objectness branch is introduced.

YOLO proposes grading each anchor box with an additional objectness score, which measures the intersection over union (IOU) value of the detection box and the ground truth box. When YOLO performs detection, the actual score of the bounding box is composed of the score of both classification and objectness. This paper then introduced an objectness subsection to the detection head of the model. This branch predicts a heatmap on the feature map, that is, whether a certain point is the center point of an object of any interesting category. In this way, the supervision gradients introduced by the added branch help the model to develop more robust feature representations, which are beneficial to detection.

The objectness and multi-classification branches are independently trained and predicted. During training, the supervision information of the objectness branch is generated in the same way as the multi-classification branch, and the loss function is also the focal loss. During inference, the final bounding box score is obtained by integrating the classification score with the objectness score:(6)$$\begin{array}{cc} bbox_{score} & =bbox_{cls}\times F\left(bbox_{obj}\right), \end{array}$$(7)$$\begin{array}{cc} F(x) & =\left\{\begin{array}{c} 1,\\ x, \end{array}\right.\begin{array}{cc} \; & x>0.5,\\ \; & else. \end{array} \end{array}$$

F(x) is the preprocessing function of the objectness score. Pixels with lower values in the objectness heatmap will provide supervision information to the multi-class heatmaps to reduce false positives. Experiments show that the added objectness branch contributes to improving accuracy while adding very little to computational burden.

### 3.5. Loss Function

This paper gives the following definition on the overall loss function to fit the model:(8)$$L_{det}=L_{cls}+\lambda_{size}L_{size}+\lambda_{off}L_{off}+\lambda_{obj}L_{obj},$$
where $L_{obj}$ is objectness loss, $L_{cls}$ is multi-classification loss, both of them are defined as focal loss:(9)$$L_{cls/obj}=\frac{-1}{N}\sum_{xyc}\begin{array}{c} \begin{array}{cc} {\left(1-{\hat{Y}}_{xyc}\right)}^{\alpha }\log\left({\hat{Y}}_{xyc}\right) & if\;Y_{xyc}=1, \end{array}\\ \begin{array}{cc} \begin{array}{c} ({1-Y_{xyc})}^{\beta }{\left({\hat{Y}}_{xyc}\right)}^{\alpha }\log\left(1-{\hat{Y}}_{xyc}\right) \end{array} & other \end{array}wise. \end{array}\;\;\;$$

${\hat{Y}}_{xyc}$ are ground truth heatmaps. The heatmap values around the object center point are subject to a two-dimensional Gaussian distribution, the distribution variance and radius are determined by the object scale. $Y_{xyc}$ are predicted heatmaps. In the multi-classification branch, *c* represents the amount of interested categories, while in objectness branch, c is 1. *N* refers to the amount of center points. The scale branch as well as the offset branch are trained with L1 loss, corresponding to $L_{size}$ and $L_{off}$:(10)$$\begin{array}{cc} L_{off} & =\frac{1}{N}\sum_{p}\left|{\hat{O}}_{\tilde{p}}-\left(\frac{p}{R}-\tilde{p}\right)\right|, \end{array}$$(11)$$\begin{array}{cc} L_{size} & =\frac{1}{N}\sum_{k=1}^{N}\left|\hat{S}-s\right|, \end{array}$$
where ${\hat{O}}_{\tilde{p}}$ is predicted offset and $\hat{S}$ is predicted scale, which can be defined as $\hat{S}=\left(\hat{W},\hat{H}\right)$, s=(w,h) is ground truth scale, *p* and $\tilde{p}$ represent the coordinate of the center point on the network input, as well as on the heatmaps with a down-sampling rate of *R*, respectively.

The hyper-parameter α is designed to be to two while β of focal loss is designed to be four. In addition, we set the loss coefficients $\lambda_{size}$ and $\lambda_{off}$ to 0.1 and 1. The above settings follow the CenterNet. We set $\lambda_{obj}$ to 0.5 through experiments.

### 3.6. Experiment

Massive experiments were then conducted on the UAV image datasets to verify whether the method we put forward is effective or not. The metric for evaluation includes the most commonly used precision and the amount of computation and parameters that are particularly important in resource-constrained applications.

#### 3.6.1. Datasets

(1)Visdrone-2018-VIDThe Visdrone-2018-VID dataset [40] contains 96 video clips taken by the drone with resolutions varies from 1344 × 756 to 3840 × 2160. The training set contains 56 clips, with 24,201 pictures in total, the validation set contains seven clips, with 2819 pictures in total, and the test set contains 33 clips, 12,968 pictures in total. The videos were recorded at various places withf similar surroundings. The annotated boxes were divided into ten categories, namely pedestrian, person, car, van, bus, truck, motor, bicycle, awning-tricycle and tricycle. Specifically, pedestrians and people are treated as different categories: a standing or walking man will be classified as a pedestrian; a man in other positions will be sorted to be a person. In our experiment, the training set as well as the validation set were utilized to train and test the model, respectively.(2)UAVDT-DETThe UAVDT-DET dataset [41] consists of 50 video clips with a fixed resolution of 1024 × 540, which are shot with a UAV platform at different places in cities. Thirty of the video clips were set to be the training set with 24,143 pictures in total, and the testing set contained 20 clips, with 16,592 pictures in total. The annotated boxes were divided into three categories, namely car, truck and bus. The other two clips were set apart to test the results. In our experiment, we used the Visdrone-2018-VID dataset to perform an ablation study to examine the validity of each part in the model. Meanwhile, the model we proposed is compared with the baseline approaches on the Visdrone-2018-VID and UAVDT-DET datasets.

#### 3.6.2. Metric

(1)AccuracyWe apply mean average precision (mAP) to assess the accuracy of the object detection algorithm, which averages the average precisions (APs) in various categories and the APs are calculated by precision-recall curves. The following equations define *precision* and *recall*:
(12)$$\begin{array}{cc} Precision & =\frac{TP}{TP+FP}, \end{array}$$
(13)$$\begin{array}{cc} Recall & =\frac{TP}{TP+FN}. \end{array}$$*TP*, *FP* and *FN* refer to the number of true positives, false positives and false negatives, respectively. True positives and false positives are determined by the *IOU* between the predicted box and the ground truth box in the same category: If the *IOU* is greater than a certain threshold, the detection box is true positive, otherwise it is false positive. Meanwhile, a ground truth box without matching any predictions will produce a false negative. *IOU* is defined as:
(14)$$IOU=\frac{B_{pr}\cap B_{gt}}{B_{pr}\cup B_{gt}},$$
where $B_{pr}$ and $B_{gt}$ represent the predicted box as well as the ground truth box, respectively. This paper sets the *IOU* level to 0.5 according to the Pascal VOC guidelines. The possible values of Recall range from 0 to 1. We produced a coordinate system by setting recall and precision as the x and y axis, respectively, which altogether formed a precision-recall curve. The area between the curve and the coordinate axises of each category is the AP of the algorithm in that category.(2)Model ComplexityThe metrics for evaluating the complexity of the CNN-based algorithm are the amount of computation and the parameters. When the CNN model performs forward inference, the amount of computation determines the time complexity, that is, the time required for obtaining the detection results, and the number of parameters determines the space complexity, that is, the capacity of storage medium required. Generally, floating point operations (FLOPs) are used to evaluate computational cost, and the parameter size is obtained by counting the total weights of the network. In the experiment, we also adopted these metrics.

#### 3.6.3. Training Details

Pytorch 0.4.1 was employed to run our method. All models in our experiment were trained and tested using a single NVIDIA TITAN RTX GPU with 24 GB RAM.

(1)Baseline MethodsWe chose CenterNet, Tiny YOLOv3, Pelee, and SSD for comparison. The specific training settings were as follows: For CenterNet, we trained the model using 16 as the lot size and 0.005 as the original learning rate for 180 epochs, with the learning rate decreasing by 10 each time at 90, 120 as well as 140 epochs. The weight decay was 0.0001, the momentum was 0.9, and the input size was 512 × 512. For Tiny YOLOv3, we trained the model using 32 as the lot size and 0.001 as the original learning rate for 150,000 steps, with the learning rate decreasing by 10 each time at 80,000 as well as 120,000 steps. The input size was 416 × 416. For Pelee, we trained the model using 32 as the lot size and 0.005 as the original learning rate for 150,000 steps, with the learning rate decreasing by 10 each time at 40,000, 80,000 as well as 120,000 steps. The input size was 304 × 304. For SSD, we trained the model using 32 as the lot size and 0.005 as the original learning rate for 120,000 steps, with the learning rate decreasing by 10 each time at 80,000 and 100,000 steps. The input size is 512 × 512.The weight decay was 0.0005 and momentum was 0.9 for Tiny YOLOv3, Pelee and SSD.(2)Our MethodThe model was trained using 16 as the lot size and 0.005 as the original learning rate for 180 epochs, with the learning rate decreasing by 10 each time at 90, 120 and as well as 140 epochs. The learning rate we propose, and the one in baseline approaches, both start from the $10^{-3}$ level, and decrease by 10 each time when the loss curve stops dropping. Moreover, the training epochs of these methods are sufficient and similar in size to maintain a fairer comparison. Furthermore, other training hyperparameters keep their original setting of implementation.

## 4. Results

### 4.1. Evaluation of Lightweight Backbone

The use of lightweight backbone may damage the detection accuracy. MobileNet is a type of classification network aiming to equilibrate accuracy and speed of mobile terminals properly. In our work, MobileNet is selected as the backbone network.

For the purpose of verifying the efficacy of MobileNet as a backbone network, we compared MobileNetV2 and MobileNetV3 [42] with ResNet18 for experiments, as is shown in Table 3. It can be found that the efficient architecture of MobileNetV2 as a backbone network does not damage the accuracy. Even if MobileNetV3 is more efficient than MobileNetV2, its mAP is lower. The use of MobileNetV2 reduces the amount of computation and number of parameters by 31% and 36.7% and the use of MobileNetV3 reduces the amount of computation and number of parameters by 33.6% and 49.9%. Both of them are more efficient than ResNet18.

### 4.2. Evaluation of Simultaneous Upsampling and Detection Module

For demonstrating the effectiveness of the simultaneous upsampling and detection module, we implemented the upsampling and detection functions through a shared single 1 × 1 convolutional layer and periodic shuffling; specific settings are shown in Table 1. We used MobileNetV2 as backbone for the experiment. Meanwhile, we tried to remove the last 1 × 1 convolutional layer of the MobileNetV2 so that the input dimension of the simultaneous upsampling and detection module is cut down, which further cuts down the quantity of computation and parameters. Table 4 shows the experimental results in detail. According to Table 4, the application of a whole simultaneous upsampling and detection module can help the network run more easily. Compared to the structure with the MobileNetV2 backbone, as well as the transpose convolution and independent detection head in Table 3, the simultaneous upsampling and detection module saved the computational costs by 87.5%, reducing the parameter size by 66.3% and without reducing accuracy. At the same time, we found that removing the last 1 × 1 convolutional layer of the backbone network, on the one hand, makes the computation of detection head easier, and on the other hand makes the detection more precise. Considering both the complexity and the detection accuracy, we chose to use the revised MobileNetV2 backbone network that removed the last 1 × 1 convolutional layer.

### 4.3. Evaluation of Objectness Branch

Based on the experiments in the above two subsections, we added an objectness branch to the network with the revised MobileNetV2 backbone and simultaneous upsampling and detection module, and demonstrate the contribution of objectness branch to detection accuracy, as is shown in Table 4 and Table 5. Table 5 shows that the optimal weight of objectness loss is 0.5. Table 4 shows that the objectness branch contributes to an improvement of 0.7 in mAP, while the increase in computational cost is only 0.3%. Figure 3 shows some subjective results of our method before and after adding the objectness branch. The visual score threshold is 0.3, and we can see that the objectness branch contributes to the reduction in false positives.

### 4.4. Comparisons with Other Detection Methods

Finally, this subsection compares our approach with the the current state-of-the-art lightweight or anchor-free object detection networks on the Visdrone-2018-VID and UAVDT-DET datasets. The methods include Tiny YOLOv3, Pelee, and SSD, the outputs of which are provided in Table 6. For Tiny YOLOv3, as well as Pelee, the input size follows the original settings [18,22]. Though the input resolution is diverse, what we actually focus on is the relationship between model complexity and accuracy which can be indicated by mAP, FLOPs, and Parameters. According to Table 6, for the Visdrone-2018-VID dataset, our method has improved mAP by 6.4 compared with the commonly used Tiny YOLOv3, and the computational cost and parameters’ size are only 23.2% and 48.6% of Tiny YOLOv3 respectively. Compared with Pelee, due to its small input scale, it has a certain advantage in the amount of computation, but our method is superior to Pelee 1.6 mAP, and the parameter size is only 35.5% of Pelee. Compared with MobileNetV2-SSD and SqueezeNet-SSD, we have improved mAP by 2.9 and 3.8, respectively, while the model complexity is slightly reduced.

In the experiment, the proposed method can reach about 220 FPS on GPU. For UAVDT-DET dataset, experimental results also demonstrate the advantage of the method we put forward. Figure 4 shows the comparison of some subjective outputs of our approach and Tiny YOLOv3. From the above discussion, it should be noticed that our method is a lightweight object detection algorithm that is more suitable for UAV images.

## 5. Conclusions and Future Work

This paper has brought forward a lightweight anchorless object detection approach based on the prediction of focus for UAV images. Our method is appropriate for mobile applications and has a low amount of computation and parameters. The proposed network structure includes a revised lightweight backbone network based on MobileNetV2 and an efficient detection head with a sub-pixel convolution and objectness-aware structure. Experimental results demonstrate that, compared with commonly used lightweight and anchor-based object detection methods, our method has certain advantages in the field of detection precision as well as model complexity, which can effectively provide support for traffic data collection and traffic parameter estimation tasks.

Since our current work is only focused on improving model structures at present, in the future we will explore leveraging novel data augmentation methods, introducing quantization and pruning techniques to pursue higher speed and more accurate lightweight object detection networks for UAV images. In addition, deploying the proposed method on-board and further combining the detection results with ITS applications will also be our future work.

## Figures and Tables

**Figure 1 sensors-21-05656-f001:**
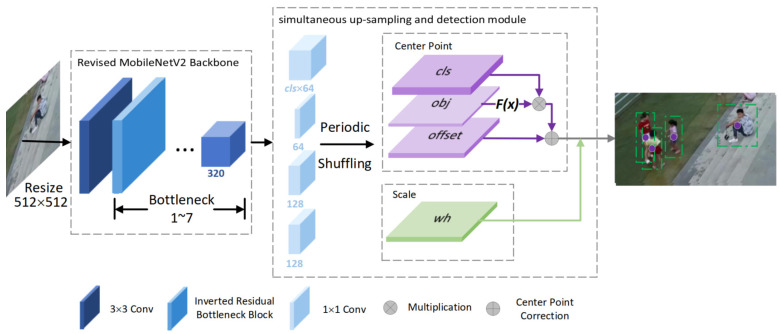
Visualization of our network architecture. Firstly, we resize the input image to fixed 512 × 512 resolution. Then, the resized image is fed into the revised MobileNetV2 backbone to obtain a feature map with 32× down-sampling. Finally, the feature maps with a large receptive field are fed into the simultaneous up-sampling and detection module. Simultaneous up-sampling and detection module integrates the function of up-sampling and detection by using the sub-pixel convolution structure. As detection head, the structure has four branches—Multi-classification branch, objectness branch, and offset branch (in purple)—that are used to determine the center of the objects of interest. Scale branch (in green) is used to determine the scale of objects of interest, that is, width and height.

**Figure 2 sensors-21-05656-f002:**
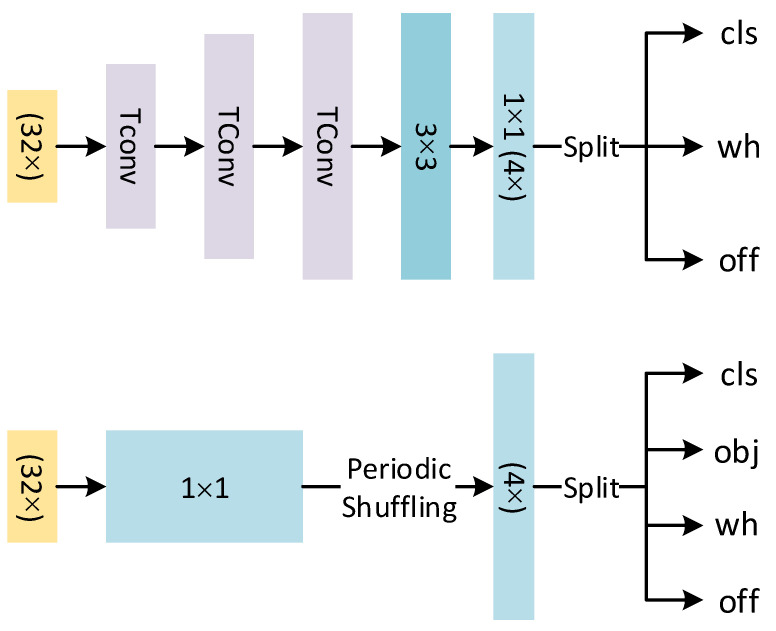
(**Top**) The counterpart of our simultaneous up-sampling and detection module in CenterNet. Tconv means transpose convolution. (**Bottom**) Our simultaneous up-sampling and detection module. The input features of 32× down-sampling are upsampled and predicted through the above structure.

**Figure 3 sensors-21-05656-f003:**
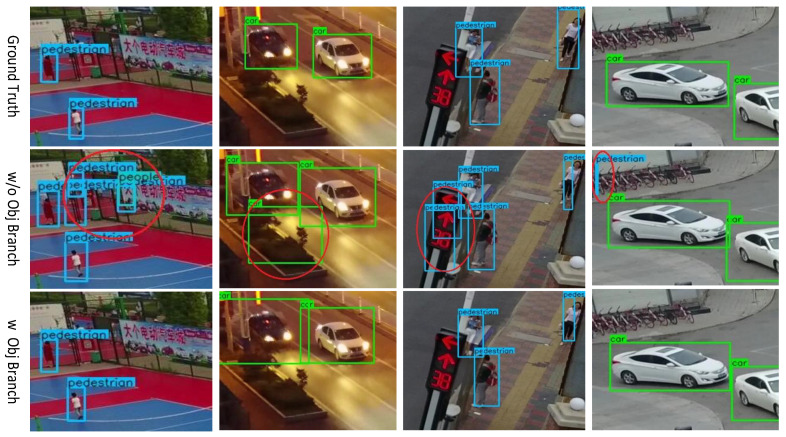
Some examples of visualizing the impact of the objectness branch. In each comparative example, the top picture shows the ground truth, the middle picture shows the results without objectness branch, the bottom picture shows the results with the objectness branch.

**Figure 4 sensors-21-05656-f004:**
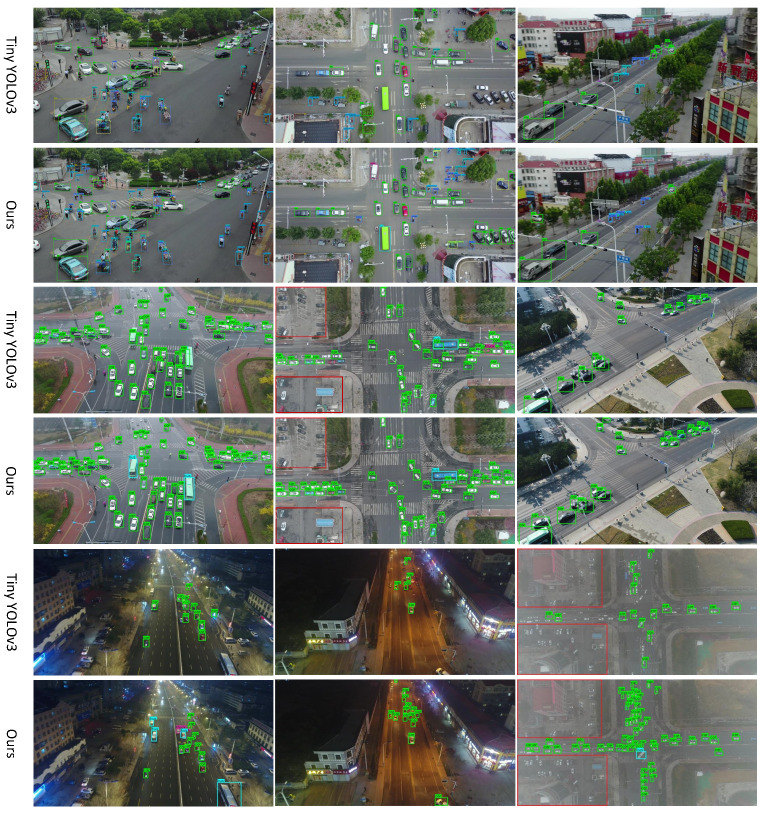
Some visualization results of our proposed method and Tiny YOLOv3. The first two lines of images are from the validation set of Visdrone-2018-VID; the last four lines of images are from the testing set of UAVDT-DET. In each comparative example, the top picture shows the results of our method, the bottom picture shows the results of Tiny YOLOv3. The areas inside the red rectangles have no annotations, so we did not detect these areas.

**Table 1 sensors-21-05656-t001:** Details of Our Network Architecture.

Layer	Configuration	Output
Backbone		
Input	—	512 × 512 × 3
Conv2d	3 × 3 × 3 × 32 s=2	256 × 256 × 32
Bottleneck	t=1 c=16 n=1 s=1	256 × 256 × 16
Bottleneck	t=6 c=24 n=2 s=2	128 × 128 × 24
Bottleneck	t=6 c=32 n=3 s=2	64 × 64 × 32
Bottleneck	t=6 c=64 n=4 s=2	32 × 32 × 64
Bottleneck	t=6 c=96 n=3 s=1	32 × 32 × 96
Bottleneck	t=6 c=160 n=3 s=2	16 × 16 × 160
Bottleneck	t=6 c=320 n=1 s=1	16 × 16 × 320
Head		
Conv2d	1 × 1 × 320 × [(cls + 5) × 64]	16 × 16 × [(cls + 5) × 64]
	*s* = 1	
Periodic Shuffling	*ratio* = 8	128 × 128 × (cls + 5)

The *t*, *c*, *n*, and *s* are the parameters of the inverted residual bottleneck structure [11]. *t* means the expansion factor. *c* means the number of output channels. *n* means repeated times. *s* means stride. Additionally, *cls* means the number of categories.

**Table 2 sensors-21-05656-t002:** Comparison of FLOPs and Parameters between CenterNet and our method.

Method	Model Complexity	Backbone		Head
		Feature Extractor	Up-Sampling	
CenterNet (ResNet18)	FLOPs	9.52 G	5.92 G	7.27 G
		(41.91%)	(26.08%)	(32.01%)
	Params	11.18M	41.20 M	0.44 M
		(71.45%)	(26.53%)	(2.81%)
ours	FLOPs	1.56 G	—	0.08 G
		(95.19%)		(4.81%)
	Params	1.81 M	—	0.31 M
		(85.46%)		(14.54%)

In CenterNet, the up-sampling module is composed of three layers of transpose convolution.

**Table 3 sensors-21-05656-t003:** Evaluation of simultaneous upsampling and detection module with MobileNetV2 Backbone and evaluation of the influence of the last convolutional layer in MobileNetV2 Backbone.

Backbone	mAP	FLOPs	Params
ResNet18	11.5	22.71 G	15.82 M
MobileNetV2	12.1	15.66 G	10.01 M
MobileNetV3	10.8	15.08 G	7.92 M

**Table 4 sensors-21-05656-t004:** Evaluation of simultaneous upsampling and detection module with MobileNetV2 Backbone and evaluation of the influence of the last convolutional layer on the MobileNetV2 Backbone.

Structure	mAP	FLOPs	Params
MobileNetV2+SUAD w/o obj	12.3	1.96 G	3.37 M
MobileNetV2 w/o last 1 × 1 +SUAD w/o obj	13.6	1.63 G	2.10 M
MobileNetV2 w/o last 1 × 1 +SUAD w obj	14.3	1.64 G	2.12 M

**Table 5 sensors-21-05656-t005:** Experiment to determine the weight of objectness loss.

$\lambda_{\mathrm{obj}}$	0.1	0.3	0.5	0.7	1.0
**mAP**	12.3	13.0	14.3	12.9	13.2

**Table 6 sensors-21-05656-t006:** Comparison of our method with the state-of-the-art Ligntweight or Anchor-free Methods on Visdrone-2018-vid and UAVDT-DET Datasets.

Method	Backbone	Input	mAP		FLOPs	Params
			VisDrone	UAVDT		
CenterNet	ResNet18	512 × 512	11.5	24.0	22.71 G	15.82 M
	MobileNetV2	512 × 512	12.1	24.6	15.66 G	10.01 M
Tiny YOLOv3	Tiny DarkNet	416 × 416	7.9	10.5	5.56 G	12.30 M
Pelee	PeleeNet	304 × 304	12.7	20.3	1.21 G	5.43 M
SSD	MobileNetV2	512 × 512	11.4	18.1	1.82 G	3.15 M
	SqueezeNet	512 × 512	10.5	20.7	1.76 G	2.33 M
Ours	MobileNetV2	512 × 512	14.3	26.6	1.64 G	2.12 M

## Data Availability

The data presented in this study are available on request from the corresponding author.

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
