# Peer review of "Lightweight Detection Network Based on Sub-Pixel Convolution and Objectness-Aware Structure for UAV Images"

_sensors, 2021, doi:10.3390/s21165656_

Round 1

Reviewer 1 Report

The manuscript "Lightweight Detection Network Based on Sub-pixel Convolution and Objectness-aware Structure for UAV Images" aims to improve existing approaches for object recognition from UAV data. 
The manuscript consists of 5 chapters, 4 rusinks, and 6 tables. 
1. Introduction contains basic information with problem statement (relatively low speed of high fidelity object recognition and "heavyweight" of existing models).
2.  Section 2 continues what was started in the introduction, but gives a more in-depth view of the existing approaches, their pros and cons. In this context, I don't see any sense in separating the introduction and section 2, and I recommend combining the material into one section with subsections.
3. The methodological section contains the basic data on the used neural network architecture, the performance of the peer networks and the proposed ways to improve the recognition results. Apart from numerous typos (which is typical for the whole manuscript in general), there is nothing to complain about. I ask the authors to pay attention to the use of hyphenation marks, and recommend not using them at all.
4. Section 4 contains Results.  I recommend to call the section so, as it is a requirement of the journal (see Instructions for Authors).  Everything before section 4.4. should be moved to the section Materials and Methods.  The section contains results of comparison of obtained results with analogues, detailed figures with indication of typical errors and error estimation metrics. Figure 4 should be moved to this section.
In general, apart from the need to correct the structure of the manuscript and typos, there are no other complaints about the text and its content.

Author Response

Response Letter

We have recently received reviews for the fore-mentioned letter. We have addressed each remark and corrected the paper based on the comments in the review.

We would like to thank you and the reviewers for the very prompt reviews of our paper and for the valuable and detailed suggestions. We believe the revised paper manuscript could answer all the questions/problems raised by the reviewers.

We thank the Reviewers, the Associate Editor and the Editor-in-Chief for their valuable suggestions toward improving our work as well as for their patience with this revision all along. We have modified the manuscript based on the reviewers’ comments, which definitely help to substantially improve our work. In this letter, we provide responses to all the questions raised by the Reviewers and incorporate all the changes as much as possible in the new version of the manuscript. The main changes are highlighted in blue in the manuscript. We believe this version is in a much better shape and hope it is satisfactory for publication in this reputed journal.

Reviewer: 1

Comments to the Author

The manuscript "Lightweight Detection Network Based on Sub-pixel Convolution and Objectness-aware Structure for UAV Images" aims to improve existing approaches for object recognition from UAV data.

The manuscript consists of 5 chapters, 4 rusinks, and 6 tables.

  1. Introduction contains basic information with problem statement (relatively low speed of high fidelity object recognition and "heavyweight" of existing models).
  2. Section 2 continues what was started in the introduction, but gives a more in-depth view of the existing approaches, their pros and cons. In this context, I don't see any sense in separating the introduction and section 2, and I recommend combining the material into one section with subsections.

Reply:Thank you for the review. We adjusted the content of the introduction, focusing on the research background. The related work part focuses on the analysis of research status. So now these two parts have their own focus.

  1. The methodological section contains the basic data on the used neural network architecture, the performance of the peer networks and the proposed ways to improve the recognition results. Apart from numerous typos (which is typical for the whole manuscript in general), there is nothing to complain about. I ask the authors to pay attention to the use of hyphenation marks, and recommend not using them at all.

Reply:Thank you for the review. These typos have been corrected in the new version.

  1. Section 4 contains Results. I recommend to call the section so, as it is a requirement of the journal (see Instructions for Authors). Everything before section 4.4. should be moved to the section Materials and Methods.  The section contains results of comparison of obtained results with analogues, detailed figures with indication of typical errors and error estimation metrics. Figure 4 should be moved to this section.

Reply: We follow this suggestion and have adjusted the structure of the last two parts of the article

In general, apart from the need to correct the structure of the manuscript and typos, there are no other complaints about the text and its content.

Reviewer 2 Report

  1. This paper is desperate. The authors need to take their submission seriously, there are many redundant “-“ in the manuscript, such as 21 lines "traf-fic", 22 lines "monitor-ing", 23 lines "cover-age ", 27 lines "traf-fic", 38 lines "de-tection", 40 lines "al-gorithms ", 42 lines "de-polying", 43 lines "be-cause", 44 lines "lim-ited", and line 47 "in-telligence", line 50 "devel-ops".... I think the authors copied it from somewhere. 125 lines "mwanwhile" means what? 291 lines “p and perepresent” is missing a " ", 433 “used lightweight” is missing a " ", line 435 “whichcan effectively” is missing a " ".
  2. Line 60 says that the method in this paper is less computationally and parametrically intensive than the Centernet with ResNet, it is not fair, you should compare the CenterNet with MobileNetV2 with your method in the experiments.
  3. Line 185 says, “The outputs of the four branches are in same spatial size (4x downsampling of input)”. Why not use the features generated by the backbone that is 1/4 the size of the input? This does not make sense.
  4. Line 198, the original image is resized to 512*512 and then detected. The result of detection is rescaled to the original image size. So the error of detection will be iterative. And those errors will affect the performance of detection. I think there is a fatal problem with this paper.
  5. Line 249, “periodic shuffling” first appeared in "Real-Time Single Image and Video Super-Resolution Using an Efficient Sub-Pixel Convolutional Neural Network" published in 2016. What the difference between your approach and the 2016 ones? It is not clear.
  6. I can not understand Eq.9.
  7. Line 344. The experiments are not representative. You should compared your method with the latest method, such as YOLO V5, CPNDet, SaccadeNet, CentripetalNet, RepPoints and etc.
  8. The result in the middle of the fourth line of Figure 4 is not reasonable. The size of cars is already very small after resizing the large image to 512*512. If the features are 32x downsampling, the car may not be detected. YOLOv3 can see the cars because it uses three different scales of feature maps, including 8x downsampling, 16x downsampling and 32x downsampling.
  9. The computation of loss about the objects’ height and width is not clear.

Author Response

Response Letter

We have recently received reviews for the fore-mentioned letter. We have addressed each remark and corrected the paper based on the comments in the review.

We would like to thank you and the reviewers for the very prompt reviews of our paper and for the valuable and detailed suggestions. We believe the revised paper manuscript could answer all the questions/problems raised by the reviewers.

We thank the Reviewers, the Associate Editor and the Editor-in-Chief for their valuable suggestions toward improving our work as well as for their patience with this revision all along. We have modified the manuscript based on the reviewers’ comments, which definitely help to substantially improve our work. In this letter, we provide responses to all the questions raised by the Reviewers and incorporate all the changes as much as possible in the new version of the manuscript. The main changes are highlighted in blue in the manuscript. We believe this version is in a much better shape and hope it is satisfactory for publication in this reputed journal.

Reviewer: 2

Comments to the Author

  1. This paper is desperate. The authors need to take their submission seriously, there are many redundant “-“ in the manuscript, such as 21 lines "traf-fic", 22 lines "monitor-ing", 23 lines "cover-age ", 27 lines "traf-fic", 38 lines "de-tection", 40 lines "al-gorithms ", 42 lines "de-polying", 43 lines "be-cause", 44 lines "lim-ited", and line 47 "in-telligence", line 50 "devel-ops".... I think the authors copied it from somewhere. 125 lines "mwanwhile" means what? 291 lines “p and perepresent” is missing a " ", 433 “used lightweight” is missing a " ", line 435 “whichcan effectively” is missing a " ".

Reply:Thank you for the detailed review. These typos have been corrected in the new version.

  1. Line 60 says that the method in this paper is less computationally and parametrically intensive than the Centernet with ResNet, it is not fair, you should compare the CenterNet with MobileNetV2 with your method in the experiments.

Reply: Centernet is not designed for lightweight object detection and takes resnet for the down-sampling part. This paper is aimed to improve centernet to make it more suitable for hardware.so we compared this method with the origin centernet. At the same time, we add the comparison with centernet which takes mobileNetv2 as backbone in table 6.

3.Line 185 says, “The outputs of the four branches are in same spatial size (4x downsampling of input)”. Why not use the features generated by the backbone that is 1/4 the size of the input? This does not make sense.

Reply: The structure of Centernet is similar to encoder-decoder. The 1/4 feature map obtained in down-sampling lacks deep high-dimensional features, while the feature map obtained after upsampling incorporates high-dimensional features. So it is better to make detection on the feature map after upsampling.

4.Line 198, the original image is resized to 512*512 and then detected. The result of detection is rescaled to the original image size. So the error of detection will be iterative. And those errors will affect the performance of detection. I think there is a fatal problem with this paper.

Reply: The images need to be resized in order to match the size of the network and the error will be compensated by the offset branch. In the comparative experiment in Table 6, we also resize the image to a small size. And the results validate the effectiveness of our schemes.

5.Line 249, “periodic shuffling” first appeared in "Real-Time Single Image and Video Super-Resolution Using an Efficient Sub-Pixel Convolutional Neural Network" published in 2016. What the difference between your approach and the 2016 ones? It is not clear.

Reply: We do agree that the idea of periodic shuffling have been proposed for super-resolution reconstruction in previous works. However, the major novelty of this work is to integrate it into a new framework which has been particularly designed for the lightweight anchor-free detection network . And the network is appropriate for mobile applications.

6.I can not understand Eq.9.

Reply: We follow this suggestion and revised the Eq.9.

7.Line 344. The experiments are not representative. You should compared your method with the latest method, such as YOLO V5, CPNDet, SaccadeNet, CentripetalNet, RepPoints and etc.

Reply: Thanks for the suggestion, but the above methods are general object detection networks which were not aimed at making the frameworks more lightweight. So it may be not suitable for comparison.

8.The result in the middle of the fourth line of Figure 4 is not reasonable. The size of cars is already very small after resizing the large image to 512*512. If the features are 32x downsampling, the car may not be detected. YOLOv3 can see the cars because it uses three different scales of feature maps, including 8x downsampling, 16x downsampling and 32x downsampling.

Reply:The method we proposed performs better than Tiny YOLOv3. The reasons are as follows:

1)In the proposed model, The objects are not detected on the feature map which is downsampled at 32x, but downsampled at 4xwhich obtained by up-sampling after 32x feature map.

2) Yolov3 needs to filter the anchors by nms. When the object‘s distribution is dense, anchors will be highly overlapping, and yolov3 may not perform well. But centernet does not require nms, it may be more flexible and accurate in dense scenes.

3) The supervision gradients introduced by the added objectness branch can help the model develop more robust feature representations, which is beneficial to detection.

9.The computation of loss about the objects’ height and width is not clear.

Reply: The definition of these variables in Eq.11 are added in the new version.

Reviewer 3 Report

sensors-1319219-peer-review-v1

The manuscript “Lightweight Detection Network Based on Sub-pixel Convolution and Objectness-aware Structure for UAV Images” addresses an interesting and up-to-date subject, which adhere to Sensors journal policies and topics.

The manuscript contains original and interesting results, well fitted in the context.

In my opinion the Abstract and Introduction part can be further improved. Very late in the text it is clear what the scope and objectives of the research are, so further improvements in the first part are necessary.

Additional English improvements are necessary.

Also, Fig 4 should be moved from Conclusions to Results.

Author Response

Response Letter

We have recently received reviews for the fore-mentioned letter. We have addressed each remark and corrected the paper based on the comments in the review.

We would like to thank you and the reviewers for the very prompt reviews of our paper and for the valuable and detailed suggestions. We believe the revised paper manuscript could answer all the questions/problems raised by the reviewers.

We thank the Reviewers, the Associate Editor and the Editor-in-Chief for their valuable suggestions toward improving our work as well as for their patience with this revision all along. We have modified the manuscript based on the reviewers’ comments, which definitely help to substantially improve our work. In this letter, we provide responses to all the questions raised by the Reviewers and incorporate all the changes as much as possible in the new version of the manuscript. The main changes are highlighted in blue in the manuscript. We believe this version is in a much better shape and hope it is satisfactory for publication in this reputed journal.

Reviewer: 3

Comments to the Author

The manuscript “Lightweight Detection Network Based on Sub-pixel Convolution and Objectness-aware Structure for UAV Images” addresses an interesting and up-to-date subject, which adhere to Sensors journal policies and topics.

The manuscript contains original and interesting results, well fitted in the context.

Reply: We would like to thank the reviewer for the very prompt and positive reviews.

In my opinion the Abstract and Introduction part can be further improved. Very late in the text it is clear what the scope and objectives of the research are, so further improvements in the first part are necessary.

Reply: Thank you for the review. We adjusted the content of the introduction, focusing on the research background.

Additional English improvements are necessary.

Reply:Thank you for the review. These typos have been corrected in the new version.

Also, Fig 4 should be moved from Conclusions to Results.

Reply:Thank you for the review. The Fig.4 has been removed to Results.

Round 2

Reviewer 1 Report

The authors took into account all the comments and edited the manuscript according to the requirements of the journal.
The manuscript deserves publication after a careful final reading.

Author Response

Dear Reviewer,

We have made a final reading carefully. Thank you very much for taking your time to review this manuscript.

Best wishes.

Reviewer 2 Report

  1. For A.3. Target detection involves classification and localization. High-dimensional features are good for classifying, and low-dimensional features are good for localizing. In your method, the objects are detected on the feature map which is downsampled at downsampled at 4x, obtained by up-sampling after 32x feature map. Your method loss the origin information extracted directly from the backbone, so your answer is not convincing to me. You can consider this question in your future work.
  2. For A.4. I do not see the results of your method at small image size in Table 6.
  3. For A.5. Cite “Real-Time Single Image and Video Super-Resolution Using an Efficient Sub-Pixel Convolutional Neural Network”, because this method is not your contribution.

Author Response

Dear Reviewer,

Thank you for your valuable suggestions toward improving our work as well as for your patience with this revision all along. We have modified the manuscript based on the your comments, which definitely help to substantially improve our work.

Comments to the Author

1For A.3. Target detection involves classification and localization. High-dimensional features are good for classifying, and low-dimensional features are good for localizing. In your method, the objects are detected on the feature map which is downsampled at downsampled at 4x, obtained by up-sampling after 32x feature map. Your method loss the origin information extracted directly from the backbone, so your answer is not convincing to me. You can consider this question in your future work.

Reply:Thank you for the review. In addition to resnet, centernet in article “CenterNet :Objects as Points” also used Hourglass network and DLA-34 as the backbone, which took both low-dimensional and high-dimensional features into account. But because of their higher .complexity, centernet with resnet as backbone is more suitable for lightweight improvement although it is slightly worse than centernet with DLA-34 and Hourglass as backbones in performance.

2For A.4. I do not see the results of your method at small image size in Table 6.

Reply:Thank you for the review. In the previous reply, the sentence“we also resize the image to a small size” might be misleading. It means that the methods used for comparison also resized the input image to 512*512 or even smaller, and the results validate the effectiveness of our schemes.

3For A.5. Cite “Real-Time Single Image and Video Super-Resolution Using an Efficient Sub-Pixel Convolutional Neural Network”, because this method is not your contribution.

Reply: We follow this suggestion and cited Real-Time Single Image and Video Super-Resolution Using an Efficient Sub-Pixel Convolutional Neural Network” at reference 12, which is highlighted in blue in the manuscript.

Best wishes.